# Assessing nitrogen dioxide monitoring techniques: a comparative analysis of Sentinel-5 Precursor satellite and ground measurements in Catalonia

Danielly Garcia Santos<sup>1</sup> and Maria Eulàlia Parés<sup>2</sup>

<sup>1,2</sup>Centre Tecnològic de Telecomunicacions de Catalunya-CERCA, Parc Mediterrani de la Tecnologia, Building B4, Av. Carl Friedrich Gauss 7, Castelldefels, 08860, Barcelona, Spain

Correspondence: Danielly Garcia Santos (danielly.garcia@cttc.cat)

**Abstract.** Effective monitoring of air pollution is essential for the development of environmental and public health policies. Comprehensive air quality management requires precise tools and strategies to assess the spatial and temporal distribution of pollutants. This study investigates the correlation between nitrogen dioxide (NO<sub>2</sub>) concentrations detected by the Sentinel-5 Precursor (S5p) satellite and those measured at ground stations by Catalonia's official air quality monitoring network during 2022 and 2023. The methodology integrates satellite and surface data aligned in space and time. The relationship between both measurements is analyzed under different frameworks: (i) global, considering the entire territory; (ii) by geographic zone (urban, suburban, rural; as well as inside and outside the Barcelona Metropolitan Area (BMA)); (iii) according to the type of stations (traffic, background, industrial); and (iv) at a seasonal level, covering different quarters of the year. Statistical tools are then used to identify patterns and differences based on zones, typology, and seasonality characteristics. The results show a moderate positive correlation at global level, with r = 0.66. By zones, the analysis reveals that suburban (r = 0.66) and non-BMA zones (r = 0.67) present stronger correlations compared to urban zones (r = 0.55), traffic typology (r = 0.61) or stations located in the BMA zone (r = 0.42). Seasonally, the correlation peaks in winter (r = 0.70) and autumn (r = 0.66), periods with more stable atmospheric conditions for  $NO_2$  concentrations, while it is lowest in spring (r = 0.61) and summer (r = 0.57). These findings highlight the utility of the S5p satellite as a complement to ground-based networks in NO<sub>2</sub> monitoring, while revealing the limitations of applying a direct relationship between both types of data at the regional level and across different geographic zones.

#### 1 Introduction

Air pollution exposure imposes a significant impact on human health globally, contributing to a substantial burden of disease. It is estimated that millions of deaths and years of healthy life are lost annually due to air pollution, making it a major global health risk, comparable to unhealthy diets and tobacco smoking (World Health Organization, 2021).

To address this issue, effective air quality monitoring is crucial for public health, environmental management, and urban development policies (Balestrini et al., 2021; Verma et al., 2024; World Health Organization, 2016). Traditional monitoring systems, such as fixed ground stations, face challenges due to limited data availability and coverage. In response, there has been

a growing interest in advanced remote sensing technologies that provide a broader and more continuous view of the dispersion of atmospheric pollutants globally (e.g., Mauro and Ullo, 2023; Morillas et al., 2024a, b; Jimenez and Brovelli, 2023; Petetin et al., 2023).

Among the numerous pollutants that affect air quality, nitrogen dioxide ( $NO_2$ ) stands out for its significant impacts on health and the environment, making it the focus of this study. It is primarily emitted from combustion processes such as vehicle engines, power plants, and industrial activities. In the atmosphere,  $NO_2$  plays a role in the formation of tropospheric ozone ( $O_3$ ) and particulate matter ( $PM_X$ ), both of which have detrimental effects on human health and the environment (Filonchyk and Peterson, 2024). Gaseous  $NO_2$  can also undergo chemical reactions to produce nitric acid and nitrate particles, contributing to acid rain and affecting ecosystems (Dvorak et al., 2010). The reactions of  $NO_2$  are influenced by factors such as sunlight intensity, temperature, humidity, and atmospheric stability. Higher temperatures and intense sunlight accelerate photochemical reactions that generate  $O_3$ , especially in warm and sunny climates. Humidity can modify the concentration of  $O_3$  precursors and the efficiency of these reactions, while atmospheric stability can lead to pollutant accumulation, particularly under high-pressure and low-wind conditions (Douros et al., 2023; Han et al., 2011; Morillas et al., 2024b; Voiculescu et al., 2020).

Remote sensing techniques enable the measurement of atmospheric  $NO_2$  by detecting how these molecules absorb sunlight in the atmosphere (ESA, 2022; Veefkind et al., 2012). Several satellite missions are currently in operation, capturing data on atmospheric gases and pollutants to provide information for air quality monitoring. This landscape is evolving rapidly; satellites with high temporal coverage, such as the Geostationary Environmental Monitoring Spectrometer (GEMS), operating in East Asia, and the Tropospheric Emissions: Monitoring Pollution (TEMPO), operating in North America, are already operational but do not provide data for Europe (Kim et al., 2020; Zoogman et al., 2017). The upcoming Sentinel-4, planned for launch in the summer of 2025, will be the first geostationary air quality monitoring mission in Europe and will provide hourly data on key air pollutants (Copernicus, 2025). In the meantime, the Sentinel-5 Precursor (S5p) satellite remains the primary satellite source for air quality monitoring in Europe. S5p is widely used for regional air quality monitoring, leveraging its advanced TROPOspheric Monitoring Instrument (TROPOMI), which provides daily global maps  $(3.5 \times 5.5 \, \text{km}^2)$  of multiple species, including  $NO_2$  gases (Douros et al., 2023; Ialongo et al., 2020; Jimenez and Brovelli, 2023; Morillas et al., 2024a, b; Pinardi et al., 2020; Omrani et al., 2020; Veefkind et al., 2012).

S5p data enhance the monitoring and understanding of air pollution on a regional scale, complementing traditional air quality monitoring systems, which often have limited coverage. Using TROPOMI, S5p measures the spectral signature of sunlight after it has passed through the atmosphere and reflected back into space. These observations are processed to produce vertical column densities (VCDs) of NO<sub>2</sub>, which represent the total amount of NO<sub>2</sub> in a column of air extending from the Earth's surface to the top of the atmosphere (Choo et al., 2023). These measurements provide a comprehensive overview of the horizontal distribution of NO<sub>2</sub>, and are particularly useful in regions with scarce or non-existent ground monitoring. Nonetheless, satellite measurements may not always accurately reflect actual ground-level NO<sub>2</sub> concentrations (Dimitropoulou et al., 2020). Despite the advanced capabilities of satellite remote sensing, a key research question remains: How strongly do NO<sub>2</sub> measurements obtained by the S5p satellite correlate with ground-based monitoring data in a geographically diverse region?

In this context, Catalonia, located in the northeast of the Iberian Peninsula, constitutes an ideal case study. The region is characterized by diverse topography, land use and climate variability within a compact area. These conditions make it an ideal region to assess the correlation between satellite-derived NO<sub>2</sub> measurements and ground-based monitoring data. To address this comprehensively, the following research questions were explored: how closely do S5p satellite NO<sub>2</sub> data correlate with surface measurements from the official measurement of air pollution network in Catalonia, considering the geographic diversity; how does the correlation between satellite data and station data vary depending on the zone or station type; how does seasonality (winter, spring, summer and autumn) influence the correlation between satellite NO<sub>2</sub> measurements and surface observations in Catalonia; and what are the benefits and limitations of using S5p data to complement surface-measured NO<sub>2</sub> information, and how could they be integrated into monitoring strategies and public policies in Catalonia?

Answering these questions is essential to understanding the reliability and limitations of satellite data when applied to regional air quality studies. Although several initiatives have examined the integration of S5p data with ground-based observations, most have focused on individual urban areas or on regions with environmental and geographical contexts that differ from broader and more diverse areas like Catalonia (e.g., Hu et al., 2024; Jimenez and Brovelli, 2023; Morillas et al., 2024a, b; Petetin et al., 2023). An exception is the study conducted by Petetin et al. (2023), which included the Iberian Peninsula but did not specifically address the heterogeneity of Catalonia. To the authors' knowledge, no previous research has applied a similar methodology exclusively to the diverse landscape of Catalonia, and compared to existing studies, this research makes three key contributions: it focuses on geographic heterogeneity by analyzing Catalonia's diverse topography and land use within a compact area; it employs a layered approach, going beyond simply comparisons between Sentinel-5P data and surface measurements by analyzing correlations between zones and station typologies; and it examines seasonal variability, studying correlations based on the season of the year, both for the complete dataset and for data grouped by zones and station typologies.

From a scientific perspective, these contributions are crucial for refining air pollution mitigation strategies and public health policies. Integrating observations of the NO<sub>2</sub> tropospheric column from the S5p satellite with point measurements from the ground-based air quality monitoring network aims to identify optimal correlation strategies between these datasets to improve the accuracy of air quality assessments.

This paper is structured as follows. Section 2 provides an overview of previous studies on the use of S5p satellite data for NO<sub>2</sub> monitoring, with emphasis on data integration approaches and current methodological challenges. Section 3 offers a brief explanation of the methodology and data used in this work. Section 4 presents the obtained results and Sect. 5 discusses the findings. Section 6 outlines the conclusions and future work for this research.

## 2 Related research

Recent investigations have explored the efficacy of integrating S5p satellite data with other data sources to accurately estimate ground-level NO<sub>2</sub> concentrations. For instance, Mauro and Ullo (2023) used machine learning approaches, specifically categorical boosting, demonstrating the potential of advanced algorithms in translating satellite observations into meaningful ground-level data (Verma et al., 2024). However, their study does not address the impact of geographical variability or regions

with limited ground-based monitoring data. Similarly, Hu et al. (2024) refined these models by examining various machinelearned models for surface NO<sub>2</sub> concentration mapping, significantly improving model accuracy by incorporating domain knowledge. Despite these advancements, unresolved challenges remain, such as error propagation in satellite-derived inputs and the need for further algorithmic improvements to enhance prediction reliability.

The work presented in Morillas et al. (2024a) investigated the correlation between S5p NO<sub>2</sub> data and ground-based observations in the Community of Madrid, Spain. They found a high linear correlation coefficient between daily NO<sub>2</sub> data and ground-based measurements, which increased when considering only the data within the satellite overpass time. Expanding on this research, Morillas et al. (2024b) examined the relationship between NO<sub>2</sub> data from air quality networks and S5p with O<sub>3</sub> and meteorological variables across Spain's largest cities. Their findings underscore the complex interactions between various pollutants and environmental factors, further emphasizing the need for integrated approaches in urban air quality management.

In addition, Petetin et al. (2023) explored the capability of the TROPOMI instrument to analyze spatiotemporal variations of  $NO_2$  as a function of land use, highlighting anthropogenic factors that influence pollutant concentrations in observations from the Iberian Peninsula. Nevertheless, more information is required to better define the profile of the observations as a function of land use, especially in regions with geographical heterogeneity and varied human activities.

Moreover, Jimenez and Brovelli (2023) estimated NO<sub>2</sub> concentrations at ground level in Milan using S5p satellite images and ERA5 meteorological data. Their model, which offered a cost-effective solution for air quality assessment in low- and middle-income countries, achieved a significantly lower Root Mean Square Error (RMSE) than the standard deviation of real measurements, indicating precise air quality monitoring from space. However, their study did not fully explore the applicability of such methods in regions with significant geographical diversity or under varying atmospheric conditions.

Collectively, these studies underscore the critical role of satellite data in enhancing the spatial resolution of air quality monitoring systems. While they reveal the potential of integrating satellite data with ground-based observations and advanced modeling techniques, they also highlight important limitations, particularly in addressing geographical heterogeneity and the accuracy of surface-level estimates. These gaps underscore the necessity for further research to refine methodologies and extend their applicability by incorporating geographically diverse areas, measurements across different zones, station typologies and seasons, providing information to optimize air quality assessment models in complex regions.

#### 3 Data and methodology

95

This section provides detailed information on the data and methodology of this study. The data from the S5p satellite and the surface data collected by the official air quality monitoring network in Catalonia are presented. Additionally, it provides an overview of the study area in Catalonia, highlighting the Air Quality Zones (AQZs) and the characteristics of the station network. The section ends with a short description on the mathematical models implemented to examine the correlation between air quality data.

# 3.1 Study area


The study focuses on Catalonia, a region in northeast Spain with an area of approximately 32,108 km<sup>2</sup>. Its geographic diversity, illustrated in Fig. 1, which includes mountains over 3,000 meters high, plains, and coasts, plays a significant role in the region's air quality (Gencat, 2022a, b; Idescat, 2020).

Catalonia presents a variety of landscapes and climates that influence the dispersion and concentration of atmospheric pollutants. Key geographic features include the Pyrenees, the Central Catalan System, and the Lleida Plain. Climatically, most of Catalonia experiences a mediterranean climate, while the Pyrenees have a mountainous climate characterized by colder and wetter conditions (Gencat, 2022b). The Pyrenees influence air currents and can act as a barrier to pollutant dispersion. The Central Catalan System, separated by the coastal plain, presents a relief that affects air circulation and pollutant dispersion, creating local microclimates. Catalonia also has a significant urban center, Barcelona, whose surrounding areas are known as the Barcelona Metropolitan Area (BMA), a region of considerable demographic and economic impact. In the interior, Lleida's extensive agricultural activity contributes to thermal inversion conditions. Along the coast, sea breezes can help disperse pollutants in coastal areas, but they can also transport them inland.

Other factors such as hydrography and anthropogenic influences also play a significant role in air quality. Major rivers such as the Ebro, Ter, Llobregat, and Besós are key features of the region. These river valleys frequently experience thermal inversion phenomena during winter, trapping pollutants close to the ground and affecting nearby populations.

## 3.2 Ground air quality data source

The ground data for this study are obtained from the *Xarxa de Vigilància i Previsió de la Contaminació Atmosfèrica* (XVPCA) network of Catalonia, which provides near-real-time (NRTI) air quality data across the study area. The XVPCA network operates monitoring stations that measure various pollutants, including NO<sub>2</sub>, PM<sub>10</sub>, PM<sub>2.5</sub>, O<sub>3</sub>, carbon monoxide (CO), and sulfur dioxide (SO<sub>2</sub>). These stations are strategically distributed across different geographic zones and typologies, providing hourly data and comprehensive coverage of the region's air quality (Gencat, 2024a).

The XVPCA network follows the European Standard EN 14211:2012 for the measurement of  $NO_2$  concentrations, which defines the chemiluminescence method for determining  $NO_2$  and  $NO_x$  levels (Gencat, 2024a). This standardized approach ensures typical uncertainties of approximately  $\pm 10\%$  under standard operating conditions (CEN, 2012).

## 3.2.1 Air quality zones in Catalonia

Catalonia is divided into several Air Quality Zones (AQZs), established to monitor and manage atmospheric pollution levels across the region. The aim of these zones is to be representative of the entire territory due to a combination of geographic, demographic, and economic activity criteria (Gencat, 2024b). The *Generalitat de Catalunya* regulates these zones to ensure compliance with air quality standards set by the European Union and national legislation. Figure 1 illustrates the geographical features of Catalonia and the distribution of air quality monitoring stations in urban, suburban, and rural zones. The detailed

specifications of these stations—including their zones, typology, measured pollutants, technology, and equipment—are provided in the Appendix A of this paper, offering further insights into the characteristics of the XVPCA network.

Figure 1. Map showing the geographical features of Catalonia and the distribution of air quality monitoring stations.

## 3.3 NO<sub>2</sub> satellite data source



The S5p satellite is part of the European Union's Copernicus Earth observation programme and it is the first Copernicus mission specifically dedicated to atmospheric monitoring. The S5p data, freely accessible through the Copernicus platform, are essential for global air quality monitoring (Veefkind et al., 2012). These data, after being corrected, are provided in Network Common Data Form (NetCDF), a widely used format for storing and sharing multidimensional scientific data (ESA, 2022).

S5p provides daily and global data on various atmospheric pollutants, including  $O_3$ , CO,  $NO_2$ ,  $SO_2$ , methane  $(CH_4)$ , formaldehyde (HCHO) and aerosols (AOD) (Veefkind et al., 2012). It uses the TROPOMI instrument to collect these data, covering a width of 2,600 kilometres, allowing the entire planet to be mapped in a 24-hour period (ESA, 2024; van Geffen et al., 2022). Of the different data provided by S5p on  $NO_2$ , in this study, the band containing the information on the tropospheric column of  $NO_2$  has been specifically used. This field provides the total amount of  $NO_2$  present in the troposphere column, which is the most relevant atmospheric region for air quality monitoring, as it reflects the direct impact of  $NO_2$  at ground level (see Fig. 2).

Datasets downloaded directly from the Copernicus platform (Level 2 data) have a native spatial resolution of approximately  $3.5 \times 5.5 \,\mathrm{km^2}$  (ESA, 2022). Based on mission requirements and validation reports, the random precision (1- $\sigma$ ) of the tropo-

spheric  $NO_2$  column is  $0.7 \times 10^{15}$  molec cm<sup>-2</sup>, while the systematic uncertainty (bias) ranges from 25% to 50% (ESA, 2015). In our study, however,  $NO_2$  values were extracted from the Sentinel-5P Level 3 dataset using the Google Earth Engine (GEE) platform. These Level 3 data are provided on a regular  $0.01^{\circ} \times 0.01^{\circ}$  grid (approximately  $1.1 \times 0.85$  km over Spain), generated by applying the harpconvert tool to the Level 2 product after filtering out pixels with qa\_value < 0.75 (GEE, 2019). This process remaps the data spatially but does not recalculate or propagate the precision band, so the official Level 2 uncertainties are retained.



It is important to note that these data can occasionally show negative values, an observation that has been documented in previous studies (Jimenez and Brovelli, 2023; Finch et al., 2022; Douros et al., 2023; Tonion and Pirotti, 2022). These negative values in tropospheric NO<sub>2</sub> columns can arise due to uncertainties inherent to the data retrieval process, which involves separating the tropospheric component column from the total NO<sub>2</sub> column by subtracting the estimated stratospheric contribution using Differential Optical Absorption Spectroscopy (DOAS) techniques (ESA, 2022; Veefkind et al., 2012). Since this separation process can be subject to errors, especially under very low NO<sub>2</sub> concentrations, negative values are generally considered to be retrieval artifacts or instrumental noise rather than physically meaningful concentrations (Jimenez and Brovelli, 2023; Finch et al., 2022; Douros et al., 2023). In particular, these cases are more likely to appear in regions with minimal NO<sub>2</sub> concentrations or complex atmospheric conditions, being within the typical instrument uncertainty range.

**Figure 2.** Tropospheric Nitrogen Dioxide column over Catalonia, measured by the Sentinel-5 Precursor satellite during the period 11-17 December 2023. Values are expressed in  $\mu$ mol m<sup>-2</sup>.

The products are available in NRTI and Offline (OFFL) modes. NRTI data are typically made available within 2 to 3 hours of acquisition by the satellite (van Geffen et al., 2022), making them ideal for applications requiring almost instantaneous

information, such as air quality monitoring, early pollution alerts or environmental emergencies. On the other hand, OFFL data, typically available 1-2 days after acquisition (Verhoelst et al., 2021), offer more comprehensive processing, with more accurate atmospheric and meteorological models, making them more suitable for detailed scientific analysis and long-term climate studies.

In addition to being available for direct download from the Sentinel-5P Pre-Operations Data Hub, its datasets are also available on Google Earth Engine (GEE), a platform that allows accessing and working with historical and NRTI atmospheric data (Salmani et al., 2023). Since analysis of large volumes of NetCDF data can be computationally expensive and slow, accessing and processing through GEE provides a more efficient alternative. In GEE, S5p data can be integrated with other datasets, such as from the XVPCA network, facilitating broader and faster analysis.

## 195 3.4 Methodology





The main objective of this study is to characterize the correlation between NO<sub>2</sub> measurements obtained from the S5p satellite and those recorded by the XVPCA surface monitoring network in Catalonia. This analysis takes into account the geographical diversity of the study area, as well as the distinctive characteristics of the monitoring stations and seasonal variations. Previous studies (Morillas et al., 2024b; Jimenez and Brovelli, 2023), have explored the integration of meteorological data in air quality analysis. However, incorporating such data at this stage could mask the direct relationship we aim to establish between satellite and ground-based NO<sub>2</sub> measurements. By focusing on this direct relationship, we aim to identify the most appropriate methods to estimate ground-level air quality using satellite data in Catalonia, leaving the integration of meteorological variables for future research.

The methodology is organized into several steps. First, data preprocessing is performed, including spatial alignment of satellite pixels with ground station locations and temporal synchronization of measurement times. Next, the analysis begins with a global approach and becomes more specific, focusing on zone-based, typology-based, and seasonal aspects. Finally, statistical methods, including correlation plots, Pearson correlation coefficients (r), regression analysis, and Coefficient of variation (CV), are applied to evaluate the relationship between satellite and ground-based data.

## 3.4.1 Data preprocessing

- To achieve the objective, it is essential to accurately match satellite and ground-based data. This matching process involves spatial and temporal alignment of data from the years 2022 and 2023.
  - 1. Spatial matching: For each ground station in the XPVCA network, the corresponding satellite pixel values are assigned based on its geographic coordinates. This ensures that the data accurately reflect the same geographic conditions. In our study, NO<sub>2</sub> values were extracted from the S5p Level 3 dataset  $(0.01^{\circ} \times 0.01^{\circ}$ , approximately  $1.1 \times 0.85$  km) using GEE. Due to this relatively high resolution and the spatial distribution of monitoring stations, we did not encounter cases of multiple stations falling within the same pixel.

However, in other study areas or when using S5p level 2 data, multiple ground monitoring stations could indeed be located within the same pixel. In such scenarios, the presence of multiple stations per pixel can be valuable for exploring within-pixel variability, revealing how surface NO<sub>2</sub> concentrations can differ even within the same satellite measurement.

2. Temporal matching: The S5p satellite passes over the study area daily around 12:00 Coordinated Universal Time (UTC) (Verhoelst et al., 2021), with data sometimes being recorded as late as 13:00 UTC. Ground stations, however, record hourly data in Central European Time (CET) or Central European Summer Time (CEST), depending on the season (Gencat, 2024c). For consistency with the satellite overpass and the subsequent analysis, the time references of ground-based measurements were standardized to UTC. This process of temporal synchronization is essential for achieving closer alignment, enabling the analysis to be performed for all time slots, relating the hourly ground-based measurements with the daily satellite overpass. This synchronization reduces discrepancies and allows for robust correlations between satellite and ground-based NO<sub>2</sub> data (Verhoelst et al., 2021; Nawaz et al., 2024; Ialongo et al., 2020).

## 3.4.2 Variable-based correlation analysis

The correlation between satellite and ground-based data is analyzed globally across the region, as well as by geographical zones, typology and season. This approach allows us to assess how different environments and seasonal factors affect the relationship between satellite and ground-based measurements.

- 1. Global: An overall correlation analysis is performed on all paired data points, providing a comprehensive view of the relationship between satellite and ground  $NO_2$  levels. This study utilizes data from 65 XVPCA stations that provide  $NO_2$  measurements.
- 2. Zone-based: Stations within the study area are categorized into two groups. The first classification is based on zones—urban, suburban, and rural—according to the definitions provided by the XVPCA network (see Table 1). The second categorization divides stations into those within the BMA zone, which includes seven urban stations, and those outside the BMA zone (non-BMA), as shown in Table 1. This categorization enables us to analyze the influence of varying population densities, urban infrastructure, and environmental conditions on the correlation between satellite and ground-based data.

**Table 1.** Number of XVPCA stations by zones and typology.

|          | Zone  | Typology-based |    |            |    |
|----------|-------|----------------|----|------------|----|
| Urban    | 22    | BMA            | 7  | Traffic    | 16 |
| Suburban | 28    | non-BMA        | 58 | Background | 37 |
| Rural    | al 15 |                |    | Industrial | 12 |

3. *Typology-based:* To address the localized impact of pollution sources, such as vehicular traffic or industrial activity, which may affect the relationship between satellite and ground-level NO<sub>2</sub> measurements, correlation analyses are performed for each station typology: traffic, background, and industrial (Table 1).

4. Seasonal: To investigate the influence of seasonal variations on the correlation, the data were grouped into four quarters of the year as an approximation for dividing the data by seasons (winter, spring, summer, autumn). This approach was necessary because the use of annual data did not allow for precise alignment with the exact start and end dates of each season. This analysis aims to identify whether seasonal factors, such as temperature and sunlight intensity, impact the correlation between satellite and ground-based measurements.

#### 3.4.3 Statistical tools for correlation analysis

To comprehensively assess the relationship between satellite- and ground-based NO<sub>2</sub> measurements, it is essential to employ robust statistical tools. In this section, we describe the methods proposed to analyze the correlation between these two data sources, ensuring a complete understanding of the behavior of the data across geographic regions and timescales.

- 1. Correlation plots: These plots visually represent the relationship between satellite and ground-based NO<sub>2</sub> measurements, helping identify patterns, trends, and outliers. They provide a visual basis for assessing the linearity and differences across geographic and seasonal contexts (Boslaugh and Watters, 2008; Kutner et al., 2004).
- 2. Pearson correlation coefficient: The r quantifies the strength and direction of the linear relationship between satellite and ground measurements, ranging from -1 (strong negative correlation) to +1 (strong positive correlation). This metric complements regression analysis by assessing the overall relationship between both data sources (Boslaugh and Watters, 2008; Kutner et al., 2004).
- 3. Regression analysis: Linear regression models are used to quantify the relationship between satellite- and ground-based NO<sub>2</sub> measurements. The slope ( $\beta_1$ ) indicates changes in ground-level NO<sub>2</sub> concentrations per satellite unit, while the intercept ( $\beta_0$ ) reflects potential biases. The coefficient of determination ( $R^2$ ) measures how well the model explains the relationship between both measurements (Boslaugh and Watters, 2008; Kutner et al., 2004; Chatterjee and Hadi, 2012).
  - 4. Coefficient of variation: The CV is defined as the ratio of the standard deviation to the mean  $(CV = \sigma/\mu)$  and provides a measure of the relative dispersion of the data. In this study, it has been applied to seasonal data to compare the variability of NO<sub>2</sub> measurements over the course of the year (Boslaugh and Watters, 2008; Kutner et al., 2004).

#### 4 Results

265

270

245

250

This section presents the results of the correlation analysis between  $NO_2$  concentrations measured during 2022 and 2023 using data from the S5p satellite and ground-based measurements from the XVPCA network in Catalonia. The analysis examines the overall correlation across the entire study area and evaluates specific correlations for datasets grouped by geographical zone, station typology, and seasonality.

The initial step of the analysis involved calculating the Pearson correlation coefficient (r) between the single daily satellite data and ground-based NO<sub>2</sub> concentrations recorded at each hour. Figure 3 presents the resulting correlations for both years, highlighting how the strength of this relationship varies throughout the day. In this figure, the X-axis corresponds to measurement times, while the Y-axis represents the correlation coefficients, standardized to UTC.

**Figure 3.** Hourly Pearson correlation coefficients between NO<sub>2</sub> concentrations measured by the S5p satellite (single daily measurement) and the ground-based XVPCA network (hourly measurement) for 2022 and 2023. The data are color-coded by year, and the subpanels correspond to: (a) Global correlation, (b) Zone-based correlation: BMA, (c) Zone-based correlation: non-BMA, (d) Zone-based correlation: urban, (e) Zone-based correlation: suburban, (f) Zone-based correlation: rural, (g) Typology-based correlation: traffic, (h) Typology-based correlation: background, (i) Typology-based correlation: industrial.

The results indicate that the highest correlation between the datasets predominantly occurs at 13:00 UTC, coinciding with the approximate overpass time of the S5p satellite. A secondary peak is observed at 19:00 UTC in most zones, except in rural zones, where no secondary peaks are detected. The only exception to this pattern is the BMA zone, where correlations at 13:00 UTC and 19:00 UTC are similarly high. Although the analysis was performed for all time slots, for clarity and better understanding, this paper focuses on the results corresponding to 13:00 UTC.

## 280 4.1 Global

Before presenting the results we would like to note that in the presentation of the dispersion results and statistical calculations, the original data types have been retained: surface concentration ( $\mu g m^{-3}$ ) for ground-based measurements and column-integrated values (mol m<sup>-2</sup>) for satellite observations. This approach was chosen because the conversion from column-integrated values to surface concentration involves complex calculations that require additional assumptions about the vertical distribution of NO<sub>2</sub> within the atmospheric column. Factors such as temperature, pressure, solar radiation and local meteorological conditions significantly influence this conversion. Including these variables would introduce an additional layer of uncertainty into the analysis, potentially altering the direct relationship between the data sets.

Hereafter we present two scatter diagrams comparing the  $NO_2$  concentrations measured by the S5p satellite and by the XVPCA ground-based network (Fig. 4), along with a table containing the correlation coefficient r and regression parameters (Table 2). The global analysis of the data considers all the values measured in Catalonia during the years 2022 and 2023 without applying any filters or divisions to the data set.

Figure 4. Global scatter plots comparing NO<sub>2</sub> concentrations measured by Sentinel-5P (mol m<sup>-2</sup>) and ground-based stations ( $\mu$ g m<sup>-3</sup>) from the XVPCA network in Catalonia for 2022 (a) and 2023 (b). Each panel includes the linear regression line with its corresponding equation, the number of data points (n), the Pearson correlation coefficient (r), and the coefficient of determination ( $R^2$ ). Calculations are based on daily satellite data matched with surface NO<sub>2</sub> measurements recorded at 13:00 UTC.

**Table 2.** Pearson correlation coefficient (r), slope  $(\beta_1)$ , intercept  $(\beta_0)$ , and coefficient of determination  $(R^2)$  for NO<sub>2</sub> levels in Catalonia during 2022–2023. Calculations are based on daily satellite data matched with surface NO<sub>2</sub> measurements recorded at 13:00 UTC.

| Year | r    | $oldsymbol{eta_1}$ | $oldsymbol{eta_0}$ | $R^2$ |
|------|------|--------------------|--------------------|-------|
| 2022 | 0.65 | 3.00e-06           | 2.57e-05           | 0.43  |
| 2023 | 0.66 | 2.75e-06           | 2.84e-05           | 0.43  |

In this study, the negative values of satellite measurements were kept in the scatter diagrams and statistical calculations because the main objective is to understand the direct correlation between  $NO_2$  measurements from the S5p satellite and those recorded by the XVPCA terrestrial network in Catalonia. Keeping these values in the analyses provided us with valuable information on that correlation in urban zones.

In the global results (Fig. 4 and Table 2), the r was 0.65 in 2022 and 0.66 in 2023, indicating a moderate positive correlation between measurements made by the S5p satellite and surface stations. Table 2 shows that the  $\beta_1$  values remained close between the two years, with a value of 2.4e-06 in 2022 and 2.2e-06 in 2023, indicating that the sensitivity of satellite measurements to changes in terrestrial NO<sub>2</sub> concentrations is constant.

However, using satellite measurements in their original unit (mol m<sup>-2</sup>) does not allow a direct interpretation for the analysis of the regression line. For this reason, we present a conversion to equivalent units ( $\mu$ g m<sup>-3</sup>) to facilitate result interpretation. This conversion do not take into account the atmosphere complexity and just consider it as homogeneous. Consequently, the regression lines have been recalculated, this time using values measured by S5p converted to  $\mu$ g m<sup>-3</sup>. The results of these lines reveal, in general, a low  $\beta_1$  in the analyzed data. For the global dataset, the  $\beta_1$  value of 0.0163 indicates that, for each increase of 1  $\mu$ g m<sup>-3</sup> in the NO<sub>2</sub> concentration measured at the surface, the concentration measured by the S5p satellite only increases by 0.0163  $\mu$ g m<sup>-3</sup>. Additionally, the  $\beta_0$  of 11.5  $\mu$ g m<sup>-3</sup> suggests that even when surface measurements are zero, the satellite still predicts a positive concentration. The R<sup>2</sup> obtained is 0.43 for both years, indicating that 43% of the variance in satellite measurements is explained by surface measurements.

#### 4.2 Zone-based

A correlation analysis was performed for two groups based on zones: one based on urban, suburban, and rural zones, and the other distinguishing between BMA and non-BMA zones. This approach allowed us to assess variations in correlation patterns across different zone types.

#### 4.2.1 Urban, suburban, and rural zones

The first analysis focuses on data grouped by urban, suburban, and rural zones, as shown in Fig. 5, with the statistical results presented in Table 3.

In urban zones, moderate correlation coefficients were observed in both 2022 and 2023 (Table 3), with r values of 0.54 and 0.55, respectively. Suburban zones exhibit the strongest correlations in this dataset, with values of 0.68 in 2022 and 0.66 in 2023. Meanwhile rural zones show moderate correlations, with r of 0.58 in 2022 and 0.55 in 2023.

In terms of the regression parameters (Table 3),  $\beta_1$  values are relatively consistent across zones and years, ranging from  $2.24 \times 10^{-6}$  to  $3.21 \times 10^{-6}$ , indicating a generally stable relationship between satellite and ground-based measurements. However,  $\beta_0$  shows greater variability, with higher values in urban zones ( $3.98 \times 10^{-5}$  in 2022 and  $4.15 \times 10^{-5}$  in 2023), suggesting a systematic overestimation of NO<sub>2</sub> concentrations by the satellite in these zones when surface values are low. Conversely, suburban zones achieve the highest R<sup>2</sup> values (0.46 in 2022 and 0.44 in 2023), reflecting better alignment between satellite and ground measurements in these less complex environments.

Figure 5. Scatter plots comparing NO<sub>2</sub> concentrations measured by Sentinel-5P (mol m<sup>-2</sup>) and ground-based stations ( $\mu$ g m<sup>-3</sup>) from the XVPCA network in Catalonia for 2022 and 2023, grouped by zone type: (a) urban 2022, (b) urban 2023, (c) suburban 2022, (d) suburban 2023, (e) rural 2022, and (f) rural 2023. Each panel includes the linear regression line with its corresponding equation, the number of data points (n), the Pearson correlation coefficient (r), and the coefficient of determination ( $R^2$ ).

**Table 3.** Pearson correlation coefficient (r), slope  $(\beta_1)$ , intercept  $(\beta_0)$ , and coefficient of determination  $(R^2)$  for NO<sub>2</sub> levels in different zones (Urban, Suburban, and Rural) during 2022–2023. Calculations are based on daily satellite data matched with surface NO<sub>2</sub> measurements recorded at 13:00 UTC.

| Zone     |      | 20                 | 22                 |       | 2023 |                    |                    |       |  |  |
|----------|------|--------------------|--------------------|-------|------|--------------------|--------------------|-------|--|--|
| Zone     | r    | $oldsymbol{eta_1}$ | $oldsymbol{eta_0}$ | $R^2$ | r    | $oldsymbol{eta_1}$ | $oldsymbol{eta_0}$ | $R^2$ |  |  |
| Urban    | 0.54 | 2.53e-06           | 3.98e-05           | 0.29  | 0.55 | 2.24e-06           | 4.15e-05           | 0.31  |  |  |
| Suburban | 0.68 | 3.21e-06           | 2.16e-05           | 0.46  | 0.66 | 3.00e-06           | 2.52e-05           | 0.44  |  |  |
| Rural    | 0.58 | 2.83e-06           | 1.97e-05           | 0.33  | 0.55 | 2.73e-06           | 2.15e-05           | 0.30  |  |  |

# 325 4.2.2 BMA and non-BMA zones

The second analysis focuses on data grouped by BMA and non-BMA zones, as shown in Fig. 6, with the corresponding statistical results presented in Table 4.

**Table 4.** Pearson correlation coefficient (r), slope  $(\beta_1)$ , intercept  $(\beta_0)$ , and coefficient of determination  $(R^2)$  for NO<sub>2</sub> levels in BMA and non-BMA zones during 2022–2023. Calculations are based on daily satellite data matched with surface NO<sub>2</sub> measurements recorded at 13:00 UTC.

| Zone    |      | 20                 | 22                 |       | 2023 |                    |                    |       |  |  |
|---------|------|--------------------|--------------------|-------|------|--------------------|--------------------|-------|--|--|
| Zone    | r    | $oldsymbol{eta_1}$ | $oldsymbol{eta_0}$ | $R^2$ | r    | $oldsymbol{eta_1}$ | $oldsymbol{eta_0}$ | $R^2$ |  |  |
| BMA     | 0.47 | 2.11e-06           | 5.68e-05           | 0.22  | 0.51 | 1.86e-06           | 5.55e-05           | 0.26  |  |  |
| non-BMA | 0.67 | 3.12e-06           | 2.26e-05           | 0.45  | 0.67 | 2.88e-06           | 2.58e-05           | 0.44  |  |  |

Despite the difference in the number of monitoring stations (see Table 1) between the urban zone (22 stations) and the BMA zone (7 stations), the r results for these two zones are quite similar. In the BMA zone (Table 4), the values were slightly lower (0.47 and 0.51) for the years 2022 and 2023, respectively. Furthermore, as observed in panels (a) and (b) of Fig. 6, no negative satellite values were detected at the locations of the BMA ground-based stations.

## 4.3 Typology-based



Hereafter, we present the correlation analysis based on different station typologies: traffic, background, and industrial stations. This analysis allows us to assess how  $NO_2$  concentrations align with satellite data, depending on the type of activity in the vicinity of the stations. The results for 2022 and 2023 are illustrated in Fig. 7, which displays scatter plots and regression lines for each typology. The corresponding statistical values are summarized in Table 5.

For traffic stations, the correlation values were moderate, at 0.63 in 2022 and 0.61 in 2023. Background stations showed slightly higher correlations, with values of 0.66 in 2022 and 0.68 in 2023. Similarly, industrial stations demonstrated correlations of 0.68 in 2022 and 0.66 in 2023.

**Figure 6.** Scatter plots comparing NO<sub>2</sub> concentrations measured by Sentinel-5P (mol m<sup>-2</sup>) and ground-based stations ( $\mu$ g m<sup>-3</sup>) from the XVPCA network in Catalonia for 2022 and 2023, grouped by BMA and non-BMA zones: (a) BMA 2022, (b) BMA 2023, (c) non-BMA 2022, and (d) non-BMA 2023. Each panel includes the linear regression line with its corresponding equation, the number of data points (n), the Pearson correlation coefficient (r), and the coefficient of determination ( $R^2$ ).

**Table 5.** Pearson correlation coefficient (r), slope  $(\beta_1)$ , intercept  $(\beta_0)$ , and coefficient of determination  $(R^2)$  for NO<sub>2</sub> levels based on station typology: Traffic, Background, and Industrial during 2022–2023. Calculations are based on daily satellite data matched with surface NO<sub>2</sub> measurements recorded at 13:00 UTC.

| Typology   |      | 20                 | 22       |       | 2023 |          |                    |       |  |  |
|------------|------|--------------------|----------|-------|------|----------|--------------------|-------|--|--|
|            | r    | $oldsymbol{eta_1}$ | $eta_0$  | $R^2$ | r    | $eta_1$  | $oldsymbol{eta_0}$ | $R^2$ |  |  |
| Traffic    | 0.63 | 2.61e-06           | 2.62e-05 | 0.39  | 0.61 | 2.35e-06 | 3.01e-05           | 0.37  |  |  |
| Background | 0.66 | 3.54e-06           | 2.51e-05 | 0.44  | 0.68 | 3.24e-06 | 2.80e-05           | 0.46  |  |  |
| Industrial | 0.68 | 2.89e-06           | 2.06e-05 | 0.46  | 0.66 | 2.71e-06 | 2.32e-05           | 0.43  |  |  |

Figure 7. Scatter plots comparing NO<sub>2</sub> concentrations measured by Sentinel-5P (mol m<sup>-2</sup>) and ground-based stations ( $\mu$ g m<sup>-3</sup>) from the XVPCA network in Catalonia for 2022 and 2023, grouped by station typology: (a) traffic 2022, (b) traffic 2023, (c) background 2022, (d) background 2023, (e) industrial 2022, and (f) industrial 2023. Each panel includes the linear regression line with its corresponding equation, the number of data points (n), the Pearson correlation coefficient (r), and the coefficient of determination ( $R^2$ ).

## 340 **4.4 Seasonal**


Finally, we present in Table 6 a seasonal analysis that covers all recorded values from 2022 and 2023 and provides a detailed view of how seasonal factors—such as temperature and atmospheric conditions—affect both the r correlation and the coefficient of variation (CV) between satellite and surface measurements of NO<sub>2</sub> throughout the year. Additionally, Tables 7 and 8 summarize the regression analysis results, including  $\beta_1$ ,  $\beta_0$ , and  $R^2$ , for NO<sub>2</sub> measurements across different seasons and typologies, highlighting the seasonal variability observed in the data.

**Table 6.** Pearson correlation coefficients (r) for NO<sub>2</sub> levels across different seasonal periods in 2022 and 2023, categorized by global values, zones, and typologies. The coefficient of variation (CV) provides an additional measure of relative variability across the seasonal periods, calculated using data from both years. Calculations are based on daily satellite data matched with surface NO<sub>2</sub> measurements recorded at 13:00 UTC.

| Type         | Wii    | nter    | Spi     | ring  | Sun  | nmer | Aut  | umn  | CV    |
|--------------|--------|---------|---------|-------|------|------|------|------|-------|
| Турс         | 2022   | 2023    | 2022    | 2023  | 2022 | 2023 | 2022 | 2023 | (%)   |
| Global       | 0.70   | 0.70    | 0.57    | 0.61  | 0.55 | 0.57 | 0.63 | 0.66 | 9.46  |
| Zone-based - | Urban, | Suburb  | an, and | Rural |      |      |      |      |       |
| Urban        | 0.62   | 0.60    | 0.43    | 0.49  | 0.38 | 0.45 | 0.50 | 0.55 | 16.61 |
| Suburban     | 0.71   | 0.69    | 0.61    | 0.60  | 0.56 | 0.54 | 0.63 | 0.66 | 9.56  |
| Rural        | 0.64   | 0.62    | 0.52    | 0.45  | 0.49 | 0.42 | 0.48 | 0.51 | 14.97 |
| Zone-based - | BMA a  | nd non- | BMA     |       |      |      |      |      |       |
| BMA          | 0.66   | 0.55    | 0.32    | 0.40  | 0.23 | 0.41 | 0.46 | 0.56 | 31.03 |
| non-BMA      | 0.70   | 0.70    | 0.61    | 0.64  | 0.59 | 0.57 | 0.64 | 0.66 | 7.46  |
| Typology-bas | sed    |         |         |       |      |      |      |      |       |
| Traffic      | 0.68   | 0.66    | 0.56    | 0.54  | 0.47 | 0.56 | 0.59 | 0.63 | 11.75 |
| Background   | 0.74   | 0.70    | 0.55    | 0.64  | 0.58 | 0.59 | 0.64 | 0.67 | 10.02 |
| Industrial   | 0.69   | 0.71    | 0.63    | 0.58  | 0.59 | 0.49 | 0.65 | 0.66 | 11.28 |

The correlation coefficients show marked seasonal variability throughout the year. Overall, winter and autumn stand out with the highest correlations, reaching a value of 0.70 in both years. In contrast, during spring and summer, significant decreases are observed in all zones and typologies, with the global minimum being 0.55 in the summer of 2022 and 0.57 in 2023. This seasonal reduction is especially evident in urban and the BMA zones, where correlations dropped to 0.38 and 0.23, respectively, in the summer of 2022.

The CV supports these observations by providing a measure of the relative variability of the seasonal correlations. The urban and BMA zones showed the highest CVs, at 16.61% and 31.03%, respectively, indicating a high degree of fluctuation. By contrast, the non-BMA zone exhibited a much lower CV (7.46%), reflecting more consistent correlations with reduced variability throughout the year.

**Table 7.** Regression analysis results showing slope  $(\beta_1)$ , intercept  $(\beta_0)$ , and coefficient of determination  $(R^2)$  for NO<sub>2</sub> levels measured during Winter and Spring in 2022 and 2023, divided by global values, zones, and typologies. Calculations are based on daily satellite data matched with surface NO<sub>2</sub> measurements recorded at 13:00 UTC.

|              |            | Winter      |         |          |          |       |          | Spring   |       |          |          |       |  |
|--------------|------------|-------------|---------|----------|----------|-------|----------|----------|-------|----------|----------|-------|--|
| Type         |            | 2022        |         |          | 2023     |       |          | 2022     |       |          | 2023     |       |  |
|              | $\beta_1$  | $eta_0$     | $R^2$   | $eta_1$  | $eta_0$  | $R^2$ | $eta_1$  | $eta_0$  | $R^2$ | $eta_1$  | $eta_0$  | $R^2$ |  |
| Global       | 3.74e-06   | 2.42e-05    | 0.49    | 3.44e-06 | 2.83e-05 | 0.49  | 2.04e-06 | 3.01e-05 | 0.33  | 2.30e-06 | 2.92e-05 | 0.37  |  |
| Zone-based - | Urban, Sul | burban, and | l Rural |          |          |       |          |          |       |          |          |       |  |
| Urban        | 3.58e-06   | 3.64e-05    | 0.38    | 2.97e-06 | 4.24e-05 | 0.36  | 1.31e-06 | 4.70e-05 | 0.19  | 1.65e-06 | 4.46e-05 | 0.24  |  |
| Suburban     | 3.54e-06   | 2.38e-05    | 0.50    | 3.53e-06 | 2.65e-05 | 0.47  | 2.84e-06 | 2.20e-05 | 0.37  | 2.81e-06 | 2.37e-05 | 0.36  |  |
| Rural        | 3.17e-06   | 2.05e-05    | 0.41    | 3.15e-06 | 2.22e-05 | 0.38  | 2.50e-06 | 1.96e-05 | 0.27  | 2.36e-06 | 2.12e-05 | 0.20  |  |
| Zone-based - | BMA and    | non-BMA     |         |          |          |       |          |          |       |          |          |       |  |
| BMA          | 4.24e-06   | 4.43e-05    | 0.43    | 2.57e-06 | 5.74e-05 | 0.30  | 7.11e-07 | 6.36e-05 | 0.10  | 1.06e-06 | 5.87e-05 | 0.16  |  |
| non-BMA      | 3.45e-06   | 2.45e-05    | 0.49    | 3.52e-06 | 2.61e-05 | 0.50  | 2.58e-06 | 2.42e-05 | 0.38  | 2.75e-06 | 2.44e-05 | 0.41  |  |
| Typology-bas | sed        |             |         |          |          |       |          |          |       |          |          |       |  |
| Traffic      | 3.31e-06   | 2.00e-05    | 0.47    | 3.17e-06 | 2.78e-05 | 0.44  | 1.73e-06 | 3.33e-05 | 0.31  | 1.72e-06 | 3.60e-05 | 0.29  |  |
| Background   | 4.69e-06   | 2.07e-05    | 0.55    | 3.77e-06 | 2.94e-05 | 0.49  | 2.19e-06 | 3.12e-05 | 0.30  | 2.96e-06 | 2.67e-05 | 0.41  |  |
| Industrial   | 3.10e-06   | 2.40e-05    | 0.48    | 3.38e-06 | 2.27e-05 | 0.51  | 2.91e-06 | 1.91e-05 | 0.40  | 2.53e-06 | 2.22e-05 | 0.34  |  |

Results by station typology indicate that background stations maintained the highest correlations throughout the year, peaking at 0.74 in winter 2022 and a CV of 10.02%, suggesting stable seasonal performance. Industrial stations also demonstrated strong seasonal stability (CV 11.28%), with correlations exceeding 0.65 during the colder months. In contrast, traffic stations exhibited the greatest seasonal variability, with correlations falling to 0.47 in summer 2022 but recovering to 0.68 in winter 2022. Nevertheless, their CV of 11.75% highlights this intermediate level of variability.

The regression analysis, summarized in Tables 7 and 8, supports these findings.  $\beta_1$  and  $R^2$  were highest in winter and autumn, reflecting a stronger sensitivity of satellite measurements to ground-level NO<sub>2</sub> during these seasons. Conversely, the summer months exhibited lower values for both. The  $\beta_0$  are consistently positive, with higher values observed in urban zones during summer and autumn.

## 5 Discussion

This section discusses the results, focusing on the interpretation across the entire study area, as well as the differences by geographical zones, station typology, and seasonality. Additionally, the implications of these findings are contextualized within existing literature.

**Table 8.** Regression analysis results showing slope ( $\beta_1$ ), intercept ( $\beta_0$ ), and coefficient of determination ( $R^2$ ) for NO<sub>2</sub> levels measured during Summer and Autumn in 2022 and 2023, divided by global values, zones, and typologies. Calculations are based on daily satellite data matched with surface NO<sub>2</sub> measurements recorded at 13:00 UTC.

|              |            | Summer      |         |          |           |       |           | Autumn    |       |           |           |       |  |
|--------------|------------|-------------|---------|----------|-----------|-------|-----------|-----------|-------|-----------|-----------|-------|--|
| Type         |            | 2022        |         |          | 2023      |       |           | 2022      |       |           | 2023      |       |  |
|              | $\beta_1$  | $\beta_0$   | $R^2$   | $eta_1$  | $\beta_0$ | $R^2$ | $\beta_1$ | $\beta_0$ | $R^2$ | $\beta_1$ | $\beta_0$ | $R^2$ |  |
| Global       | 2.20e-06   | 2.83e-05    | 0.30    | 2.04e-06 | 2.70e-05  | 0.33  | 2.83e-06  | 3.00e-05  | 0.40  | 2.48e-06  | 3.43e-05  | 0.44  |  |
| Zone-based - | Urban, Sul | burban, and | l Rural |          |           |       |           |           |       |           |           |       |  |
| Urban        | 1.37e-06   | 4.54e-05    | 0.15    | 1.54e-06 | 3.88e-05  | 0.21  | 2.33e-06  | 4.96e-05  | 0.25  | 1.99e-06  | 4.97e-05  | 0.31  |  |
| Suburban     | 2.79e-06   | 2.30e-05    | 0.31    | 2.06e-06 | 2.58e-05  | 0.29  | 2.84e-06  | 2.60e-05  | 0.39  | 2.62e-06  | 3.20e-05  | 0.44  |  |
| Rural        | 2.50e-06   | 1.81e-05    | 0.24    | 2.15e-06 | 1.94e-05  | 0.18  | 2.28e-06  | 2.39e-05  | 0.23  | 2.35e-06  | 2.57e-05  | 0.26  |  |
| Zone-based - | BMA and    | non-BMA     |         |          |           |       |           |           |       |           |           |       |  |
| BMA          | 6.81e-07   | 6.35e-05    | 0.05    | 1.21e-06 | 5.06e-05  | 0.17  | 1.93e-06  | 6.83e-05  | 0.21  | 1.94e-06  | 6.38e-05  | 0.32  |  |
| non-BMA      | 2.64e-06   | 2.34e-05    | 0.34    | 2.11e-06 | 2.50e-05  | 0.32  | 2.90e-06  | 2.67e-05  | 0.41  | 2.47e-06  | 3.23e-05  | 0.43  |  |
| Typology-bas | sed        |             |         |          |           |       |           |           |       |           |           |       |  |
| Traffic      | 1.55e-06   | 3.57e-05    | 0.22    | 1.73e-06 | 2.86e-05  | 0.31  | 2.57e-06  | 3.10e-05  | 0.34  | 2.16e-06  | 3.51e-05  | 0.40  |  |
| Background   | 3.04e-06   | 2.50e-05    | 0.34    | 2.64e-06 | 2.57e-05  | 0.35  | 3.14e-06  | 3.13e-05  | 0.41  | 2.84e-06  | 3.52e-05  | 0.45  |  |
| Industrial   | 2.43e-06   | 2.05e-05    | 0.34    | 1.60e-06 | 2.42e-05  | 0.24  | 2.52e-06  | 2.37e-05  | 0.43  | 2.40e-06  | 2.79e-05  | 0.44  |  |

The consistently high correlation coefficients (r) at 13:00 UTC (Fig. 3) across most datasets and both years confirm that the strongest correlation between satellite and surface measurements tends to occur at the approximate overpass time of the S5p satellite (Verhoelst et al., 2021). However, the behavior in the BMA zone diverges from this trend, showing similarly high correlations at both 13:00 UTC and 19:00 UTC. This distinct pattern underscores the complexity of representing surface-level  $NO_2$  in densely urbanized zones, likely influenced by evening traffic emissions typical of high-density urban areas.

# 5.1 Global



The 2022 scatter plots (Fig. 4) shows a greater dispersion of the measurements when compared to the 2023 graph. This difference is partly attributed to improvements implemented in the processing and calibration of the S5p data, which have reduced the bias and increased the accuracy of the measurements (ESA, 2023). These refinements demonstrate the impact of continuous updates in satellite data processing on improving the reliability of satellite-ground correlations.

The global results indicate a moderate positive correlation between measurements made by the S5p satellite and surface stations, Table 2. This result verifies that there is a general correspondence between both data sets, although it is not strong enough to be considered as a good agreement between the data. Similar observations have been reported in previous studies. For instance, Ialongo et al. (2020) found a moderate positive correlation coefficient (r = 0.68) between TROPOMI NO<sub>2</sub> total column and ground-based measurements in Helsinki.

In contrast, the results obtained by Morillas et al. (2024a) in a study conducted in Madrid reported moderately higher correlation coefficients (r = 0.78) compared to those presented in Table 2 of this study. A notable difference between the two studies is that, while the analysis conducted here observed higher correlations in rural zones than in urban zones, the study in Madrid identified the opposite trend. Finally, in the same study, the authors agreed that the geographic distribution of monitoring stations can significantly influence the correlation coefficients obtained. These differences in results can be directly attributed to the size of the study areas. While their research focused exclusively on the city of Madrid ( $8,028 \text{ km}^2$ ), the present analysis covered the entire region of Catalonia ( $32,108 \text{ km}^2$ ). This larger area introduces variations in geography and area characteristics that influence the results.

Regression analysis, using S5p values converted to  $\mu$ g m<sup>-3</sup>, reveals a  $\beta_1$  of 0.0163 and a  $\beta_0$  of 11.5  $\mu$ g m<sup>-3</sup>. The low  $\beta_1$  suggests that variations in surface measurements are not proportionally captured by the satellite. The positive  $\beta_0$  implies a bias in the data, even when surface measurements are 0, the satellite predicts a positive value. As with r, where the results indicate a general correspondence, the R<sup>2</sup> (Table 2) obtained indicates that 43% of the variance in satellite measurements is explained by surface measurements, leaving 57% of the variance attributable to other factors not captured by ground-based monitoring. These values are essential for studies that seek to estimate surface concentrations from satellite data. In this analysis, are important for understanding how the correlation between satellite and surface data evolves in response to changes in measured air pollutant levels.

#### 5.2 Zone-based




This section analyzes how the correlation between satellite measurements and surface data varies depending on the two groups defined by zones (urban, suburban and rural) and (BMA and non-BMA). The discussion focuses on interpreting the patterns observed in the correlation coefficients and in the regression adjustments.

#### 5.2.1 Urban, suburban, and rural zones

The results in Table 3 and Fig. 5 show that the highest correlations are observed at stations classified as suburban zone (r = 0.68 in 2022, panel (c), and r = 0.66 in 2023, panel (d)), followed by rural, and finally urban zones. These correlation results are consistent with the R<sup>2</sup> values, which remain moderate—0.43 for the global results, and 0.46 and 0.44 for suburban zones in 2022 and 2023, respectively. This pattern can be attributed primarily to the intensity and distribution of emissions in each zone type. In suburban zones, the moderate NO<sub>2</sub> concentrations reflect a mix of emission sources that are more dispersed than those in strictly urban environments. These results align with previous findings highlighting the influence of emissions in less dense environments on satellite-ground consistency (Wang et al., 2022; Munir et al., 2021).

Taken together, these findings underscore the importance of emission source density, source types, and spatial representativeness of measurements when interpreting relationships between  $NO_2$  values derived from satellite data and ground stations.

The scatter plots reveal distinctive patterns across the three zone types. Rural stations present a compact cloud of points concentrated at low values, indicating that most measurements fall within the low-concentration range. In contrast, urban stations show very few points near zero, reflecting consistently high NO<sub>2</sub> levels characteristic of dense urban environments.

Suburban stations, situated between these extremes, display a more balanced distribution across mid-range values, contributing to the strongest correlations with satellite observations. These differences highlight how emission intensity and distribution shape the agreement between satellite-derived and ground-based NO<sub>2</sub> measurements.

# 5.2.2 BMA and non-BMA zones

The second analysis focuses on data grouped by BMA and non-BMA zones (Fig. 6 and Table 4). The BMA scatter plots (panels (a) and (b) of Fig. 6) reveal a markedly different data dispersion compared to all other sets. Unlike the other graphs, the BMA plots do not display points clustered around zero in either the ground-based NO<sub>2</sub> measurements (x-axis) and the satellite-derived NO<sub>2</sub> columns (y-axis). Despite the low correlation values observed, these scatter plots effectively demonstrate the ability of satellite data to capture spatial differences in metropolitan zones.

The low correlation values are consistent with findings from previous studies (Munir et al., 2021; Jamali et al., 2020), which suggest that higher anthropogenic activity, topographic complexity, and urban density—typical of the BMA zone—tend to reduce the agreement between surface NO<sub>2</sub> measurements and satellite data. One of the main challenges in air pollution data in metropolitan areas is traffic variability: within the spatial extent of a single S5p pixel, multiple types of neighborhoods can coexist (from traffic-calmed areas to high-density zones) resulting in significant spatial heterogeneity in NO<sub>2</sub> concentrations.

In contrast, the absence of negative satellite values in the zone contrasts with the findings of (Douros et al., 2023; Tonion and Pirotti, 2022), who reported negative satellite concentrations in regions with complex environmental conditions. This discrepancy may be attributed to regional factors unique to Catalonia.

Conversely, non-BMA zones exhibited patterns similar to suburban zones, a finding also highlighted by other authors when comparing the effectiveness of satellite data in different geographic environments (Wang et al., 2022; Munir et al., 2021). The regression coefficient  $\beta_1$  (Table 4) illustrates key differences between these zones, indicating that changes in NO<sub>2</sub> concentrations are more effectively captured by satellites in these less complex environments. In contrast, the lower R<sup>2</sup> values in urban and BMA zones reflect the greater complexity of these environments, suggesting a less direct relationship with satellite data.

# 5.3 Typology-based




The typology-based analysis allows for the assessment of how  $NO_2$  concentrations align with satellite data depending on the type of activity in the vicinity of the stations. (Fig. 7 and Table 5).

The Traffic data, shown in panels (a) and (b) of Fig. 7, do not exhibit negative satellite values at the monitoring station locations—similar to what is observed in the Urban and BMA datasets (Figs. 5 and 6, panels (a) and (b)). This result highlights the importance of retaining negative data measured by the S5p satellite in the analyses. Although traffic stations display the lowest correlations among typologies, they are still higher than those observed in Urban and BMA zones.

In contrast, industrial and background stations show slightly stronger correlations compared to traffic stations, which is consistent with the R<sup>2</sup> results (Table 5). These findings underline the utility of separating the analysis by station typology to better understand the factors influencing satellite-ground measurements alignment.

## 5.4 Seasonal






The seasonal analysis, summarized in Table 6, reveals significant variations in the r throughout the four quarters of the year. In general, the highest correlations are observed during the coldest quarters, winter and autumn, with the maximum values reached in the first quarter of both years. This similarity in the results in both years is consistent across most zones and typologies, and also indicates the higher correlation between satellite and surface measurements of  $NO_2$  during the coldest months of the year. During the second quarter, corresponding to spring, a slight decrease in correlation is observed across all zones and typologies, although it is not significantly pronounced. Similar trends are observed for both zone-based and typology-based stations. In the third quarter, which includes the summer months, correlation coefficients reach their lowest values.

The CV provides additional insight into the stability and variability of these seasonal correlations. Urban and BMA zones exhibit the highest CV values (16.61% and 31.03%, respectively), indicating a moderate degree of fluctuation due to environmental and anthropogenic factors, such as increased photochemical reactions. In contrast, non-BMA zone show CVs of 7.46%, highlighting more stable values across seasons.

Research shows that increased solar radiation during the summer months intensifies the NO<sub>2</sub> photolysis process (Yang et al., 2004), reducing its concentration near the ground. This seasonal effect is particularly pronounced during summer due to higher solar intensity and longer daylight hours, which enhance photochemical reactions and accelerate the conversion of NO<sub>2</sub> into other compounds. Consequently, the lower correlations observed during this period can be attributed to this intensified photolysis and the resultant variability in NO<sub>2</sub> concentrations (Douros et al., 2023; Voiculescu et al., 2020; Han et al., 2011). These seasonal dynamics help to explain the variations in correlations between satellite data and surface measurements across the different quarters analyzed.

As in the zone-based analyses for the entire year, suburban and non-BMA zones have higher correlations than urban and BMA zones, indicating a greater impact of urban-specific factors. Typology-based analysis reveals that background and industrial stations maintain relatively high correlations throughout the year, with CVs of 10.02% and 11.28%, respectively. Industrial areas reached their highest correlation in the first quarter of 2023, while traffic stations exhibited slightly greater seasonal variability (CV 11.75%), with the lowest correlation occurring in the summer months, particularly in 2022, when it dropped to 0.47 (Table 6).

The  $R^2$  results also vary seasonally (Tables 7 and 8), similar to the r coefficient, with the highest percentages of explained variance recorded in the first and fourth quarters. This trend highlights the stronger relationship between satellite and ground-based  $NO_2$  measurements during the winter and autumn months. The summer months exhibited lower values for  $\beta_1$  and  $R^2$ , suggesting greater unexplained variance and reduced comparability between datasets. The higher  $\beta_0$  observed in urban zones and during summer and autumn indicates a systematic overestimation of  $NO_2$  concentrations by the satellite.

The consistently positive  $\beta_0$  values across all areas and seasons indicate that satellite measurements capture a broader regional signal, integrating contributions from atmospheric layers above ground level. Meanwhile, variability in CV,  $\beta_1$  and  $R^2$  values highlights the influence of local environmental conditions, seasonal changes, and the spatial distribution of  $NO_2$  sources on satellite-ground comparability.

## 5.5 Overall discussion







The strongest correlations are observed at 13:00 UTC—coincident with the S5p overpass—highlighting the need for temporal alignment in any satellite-based estimation model. Correlation strength varies with geographic location and station typology: urban and BMA zones display lower r values and larger seasonal variability than suburban and non-BMA zones, reflecting their greater emission density and environmental complexity.

The diversity of conditions within a satellite pixel can influence the observed correlations. A single TROPOMI pixel may cover diverse land uses, such as urban, suburban, and rural zones, or encompass areas with both point and diffuse emissions. This heterogeneity can reduce the correlation between satellite-derived NO<sub>2</sub> values and ground measurements, particularly in complex zones such as urban and BMA zones. In contrast, more homogeneous regions, such as suburban and non-BMA zones, tend to have higher correlations also related to less variability within the pixel. In terms of typological stations, background and industrial stations record higher correlations compared to traffic stations, where local emissions and geographic heterogeneity reduce the representativeness of satellite data.

During winter and autumn, stable atmospheric conditions prevail, with colder air trapped in the lower boundary layers, limiting vertical mixing. This atmospheric stability reduces photochemical reactions, allowing NO<sub>2</sub> to accumulate near the surface and contributing more significantly to the observed tropospheric column. As a result, correlations between surface and satellite data are stronger, particularly in background stations and in suburban and non-BMA zones, where lower CVs indicate more consistent seasonal alignment. In contrast, during summer months, atmospheric instability, characterized by anticyclones and higher temperatures, enhances NO<sub>2</sub> vertical mixing. This process dilutes NO<sub>2</sub> throughout the boundary layer, increases photolysis rates, and reduces the representativeness of ground-level measurements in satellite observations. This seasonal instability introduces greater variability, as reflected in higher CVs for urban and traffic stations.

The results obtained allow us to address the research questions posed at the beginning of this study. Regarding how closely do S5p satellite NO $_2$  data correlate with surface measurements from the official measurement of air pollution network in Catalonia, the study found moderate correlations on a global scale ( $r \approx 0.65$ ) and highlighted significant geographic variability. Suburban and non-BMA zones showed the strongest correlations, suggesting that satellite data more accurately capture NO $_2$  concentrations in less dense areas. Urban and BMA zones showed lower correlations due to their complexity and the influence of localized emissions. Moreover, the analysis of the correlation between satellite data and station data depending on zone or station type revealed that background and industrial stations align better with satellite data ( $r \approx 0.66$ –0.68) compared to traffic stations ( $r \approx 0.61$ –0.63). This highlights the impact of local emissions and environmental complexity in reducing correlation at traffic monitoring sites.

Regarding the seasonal variation, the analysis showed that winter and autumn reach the highest correlations (up to r = 0.70), while summer and spring present lower values. The reduced correlation during summer is attributed to stronger photolysis processes and atmospheric variability, which decrease the representativeness of satellite observations. These findings emphasize the seasonal dependence of satellite-ground agreement and suggest a specific focus on winter data for modeling.

Finally, regarding the benefits and limitations of using S5p data to complement surface-measured NO<sub>2</sub> information, and how could they be integrated into monitoring strategies and public policies in Catalonia, the results highlight that S5p data provide valuable spatial coverage, especially in suburban zones and outside the BMA zone. These data allow for the identification of pollution trends, regions for prioritizing interventions, and the better allocation of resources for installing new ground-based monitoring stations.

## 520 6 Conclusions






This study has explored the correlation between S5p satellite data and ground-based NO<sub>2</sub> measurements in Catalonia, addressing geographical diversity, station typologies and seasonal variability. The findings confirm that satellite measurements align moderately with ground-based observations, and provide insights for areas with limited ground-based station coverage.

The degree of alignment depends on factors such as geographical context, station typology and seasonal variability. Suburban and less built-up zones show better correlations and lower variability, while denser urban zones and those with concentrated emissions present lower correlations. Station typology also influences the relationship, with background and industrial stations showing stronger alignment and more stable seasonal performance.

The impact of seasonality on correlations, both for global data and for data segmented by zone and type, is very significant and demonstrates the viability of generating a model for estimating surface NO<sub>2</sub> in Catalonia. The CV values show that urban and traffic stations, as well as the summer months, experience greater seasonal variability, which can introduce additional variance into estimation models. On the one hand, it must be taken into account that including data from months with lower correlation can introduce variance that does not represent the behavior of the pollutant. On the other hand, it reinforces the idea that it is preferable to optimize computational processing by keeping only the data from the winter months, which have the highest correlations and the lowest seasonal variability, as indicated by lower CVs. It is preferable to use data from different years rather than data from the whole year.

The results align with studies that claim that it is possible to estimate surface pollution levels from satellite data. However, this study has shown that in the region of Catalonia, a direct relationship between satellite and surface data is not fully achievable due to geographic and temporal variability. This underscores the significance of this study and the results obtained for further developing an effective surface estimation methodology for a geographically diverse area such as Catalonia.

Despite its limitations, S5p data provide significant opportunities to complement traditional monitoring networks, particularly in remote regions or areas with limited ground station coverage. The coefficient of variation offers a useful tool for assessing the stability of seasonal correlations and identifying areas where satellite data may be more reliable. Integrating satellite and surface data requires careful consideration of geographic and temporal factors to enhance reliability and utility. These findings reinforce the importance of combining multiple data sources to improve the accuracy of air quality assessments and support the development of effective public policies for pollution mitigation.

For future research, if the aim is to develop a model to estimate surface NO<sub>2</sub> from satellite data, it is advisable to consider separating the data by season or focusing on data from the winter months, which have the most stable correlations. In addition,

it is necessary to integrate atmospheric variables such as temperature and solar radiation. Although the influence of these variables is lower in winter, it is still necessary to take into account the variance of approximately 50% of the surface data that are not directly influenced by satellite data. Although high accuracy in estimating surface NO<sub>2</sub> from satellite data may not be fully achievable, this approach opens the way to a useful tool for visualizing pollution levels and raising public awareness of air quality in Catalonia.

Data availability. The Sentinel-5 Precursor (S5p) NO<sub>2</sub> data used in this study are publicly available from the Copernicus Open Access Hub (https://scihub.copernicus.eu/). The ground-based air quality monitoring data for Catalonia were obtained from the *Portal de Dades*555 Obertes Catalunya and are available at https://mediambient.gencat.cat/ca/05\_ambits\_dactuacio/atmosfera/qualitat\_de\_laire/vols-saber-querespires/descarrega-de-dades/descarrega-dades-automatiques/.

## **Appendix A: Glossary of terms**

#### A1 Zone of stations


Classification of measurement points based on their geographical location:

- Urban: measurement points located in urban areas characterized by continuous construction and urban infrastructure, including public services.
  - Suburban: measurement points located on the outskirts of a city, adjacent to or very close to a highly urbanized and populated area.
    - Rural: measurement points located in areas that are neither urban nor suburban.

# 565 A2 Typology of stations

Classification of measurement points based on the type of emission sources they monitor:

- Traffic: measurement points located in areas directly impacted by traffic emissions.
- Industrial: measurement points located in areas directly impacted by industrial emissions.
- Background: measurement points located in areas not directly impacted by traffic or industrial emissions. The air at these
   points has mixed and comes from various sources.

## A3 Measured pollutants

The primary atmospheric pollutants measured by the *Xarxa de Vigilància i Previsió de la Contaminació Atmosfèrica* (XVPCA) network:

- Nitrogen dioxide (NO<sub>2</sub>): primarily emitted by vehicular traffic and some industrial activities.
- Suspended particles (PM<sub>10</sub> and PM<sub>2.5</sub>): originating from various sources such as traffic, industry, and agricultural activities.

- Ozone (O<sub>3</sub>): a secondary pollutant formed from photochemical reactions between precursors like nitrogen oxides and volatile organic compounds (VOCs).
  - Carbon monoxide (CO): emissions from incomplete combustion in vehicles and some industries.
  - Sulfur dioxide (SO<sub>2</sub>): mainly from burning fossil fuels in power plants and some industries.

# 580 A4 Technology and equipment

Technological and operational aspects of the XVPCA network:

- Automatic sensors: use advanced technologies for continuous pollutant measurement. Data are collected in real-time and sent to a control center for analysis.
- Manual sensors: samples are collected on-site, analyzed in a laboratory, validated, and then included in the atmospheric pollution database.
  - Calibration and maintenance: XVPCA stations undergo rigorous calibration and maintenance processes to ensure data accuracy and reliability.
  - Near-real-time (NRTI) data: measurement results are published a few hours later on the Generalitat de Catalunya's website, allowing citizens and authorities to access up-to-date air quality information.

590 *Author contributions*. D.G. has defined the use case and the tests to be done, has performed the calculus, has compiled the results and has analyzed them and has written the manuscript; M. E. Parés has defined the use case and the tests to be done, has analyzed the results and has written the manuscript.

Competing interests. The authors declare that they have no competing interests.

Acknowledgements. The publications and other results were carried out with the support of the predoctoral program AGAUR-FI grants (2023 FI-1 00297) Joan Oró, funded by the Secretariat of Universities and Research of the Department of Research and Universities of the Generalitat de Catalunya and the European Social Fund Plus. This work has been carried out within the framework of the Geography doctoral program of the *Universitat Autònoma de Barcelona*. The authors would like to thank Dr. Eduard Angelats for his time and all the fruitful discussions. The authors acknowledge the support of artificial intelligence tools for linguistic revision.

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
