# Peer review of "Assessing nitrogen dioxide monitoring techniques: a comparative analysis of Sentinel-5 Precursor satellite and ground measurements in Catalonia"

_EGUsphere, 2025_

## Author Response (AR1)

**Dear Editor,**

First of all, we would like to express our sincere gratitude to you and the reviewers for the valuable comments and suggestions on our manuscript. We highly appreciate the time and effort dedicated to providing insightful feedback. We have carefully considered all the comments and have made the necessary revisions to improve the manuscript. Below, we address each point raised by the reviewers in a point-by-point manner, indicating the changes made in the manuscript.

**Reviewer 1**

- S5p - discuss difference in sampling scales.

We have added further explanation in Section 1, lines 46 and 79 of the revised manuscript.

- Implications for future satellites with better spatial, temporal resolution?

Although satellites with improved temporal coverage, such as Geostationary Environment Monitoring Spectrometer (GEMS) (East Asia Timeline) and Tropospheric Emissions: Monitoring of Pollution (TEMPO) (North America Timeline), are already operational, they do not provide data for Europe (Kim, 2020; Zoogman, 2017). The upcoming Sentinel-4, scheduled for launch in the summer of 2025, will be the first geostationary air quality monitoring mission in Europe, providing hourly data on air pollutants (Copernicus, 2025). Near-real-time data will allow for more detailed monitoring of diurnal pollution patterns. Advances in these data will help refine high-resolution air quality modelling.

Applying this study to future Sentinel-4 data, the hypothesis is that by correlating hourly data measured by both the satellite and ground stations, we will obtain correlated data throughout the day, allowing us to define pollution patterns and analyze in-depth the influence of seasonal and meteorological conditions on the data correlation.

We have expanded the discussion in the manuscript in Section 1, lines 39 to 44.

**- Describe the typical vertical variation of $NO_2$ within the tropospheric column. Has anyone been successful in relating tropospheric columns and surface level $NO_2$ ?**

 $NO_2$  exhibits variations in vertical distribution within the column depending on location, emission sources, and atmospheric conditions. Typically, the highest  $NO_2$  concentrations occur in the lower tropospheric layers (0–1 km), influenced primarily by anthropogenic emissions such as traffic and industrial activities. At higher altitudes,  $NO_2$  concentrations tend to decrease due to scattering and photodissociation processes. However, under certain atmospheric conditions, such as temperature inversions or high atmospheric stability,  $NO_2$  can accumulate in specific tropospheric layers. These vertical variations affect the relationship between satellite-derived total column measurements and surface concentrations, as satellites measure total column  $NO_2$ , while ground-based stations measure surface concentrations (ESA, 2022; Veefkind et al., 2012; Landgraf et al., 2018; Eskes et al., 2019).

Studies such as Morillas et al. (2024a), Petetin et al. (2023) and Jiménez and Brobelli (2023) have obtained promising results, with correlation coefficients ranging from approximately 0.70 to 0.90,

depending on the location, meteorological conditions, and time-averaging method. However, as discussed in lines 107-109, the geographical characteristics and spatial scales considered in those studies differ from those examined in this article. Our findings indicate that such differences significantly influence the correlation between satellite and surface measurements.

We have added further explanation in Section 1, lines 51 to 55.

- How is NO2 being measured at the ground-based monitoring network sites? Some in-situ analyzers have been shown to have interferences. How relevant are those for your study?

 $NO_2$  concentrations are measured by the *Xarxa de Vigilància i Previsió de la Contaminació Atmosfèrica* (XVPCA) using automatic in-situ monitoring stations. These stations follow standardized procedures for data acquisition, including regular calibration and validation protocols (Gencat, 2024). The resulting data are published on the Catalan open data portal approximately four hours after collection. Regarding potential interferences in in-situ analyzers, for the purposes of this study, the surface  $NO_2$  data from the official XVPCA network of Catalonia are considered both robust and reliable.

We have added further explanation in Section 3.2, lines 144 to 146.

- Line 159-165: How is the tropospheric column retrieved? Then you can better understand why the tropospheric columns can be negative.

As described in lines 37-38, the tropospheric  $NO_2$  columns used in this study are retrieved using techniques that measure the absorption of sunlight by atmospheric  $NO_2$  molecules (ESA, 2022; Veefkind et al., 2012). Specifically, this retrieval involves isolating the tropospheric component by subtracting the estimated stratospheric contribution from the total  $NO_2$  column obtained through Differential Optical Absorption Spectroscopy (DOAS). Due to uncertainties and assumptions inherent in this separation, particularly under conditions of very low  $NO_2$  concentrations, negative values can occasionally result. Several studies have documented and discussed this issue, including Jimenez and Brovelli (2023), Finch et al. (2022), and Douros et al. (2023), highlighting that these negative values are primarily retrieval artifacts or instrument noise rather than physically meaningful concentrations (lines 159-165). In practice, such negative columns tend to appear over regions with minimal  $NO_2$  or complex atmospheric conditions and remain within the instrument's typical uncertainty range.

We have added further explanation in Section 3.3, lines 176 to 183.

- Figure 2: "high" and "low" levels of NO2 not descriptive enough. Give the actual column values and units.

We have updated Figure 2 to include the actual column values and the corresponding units.

- Spatial matching: How large are the pixels? Are there pixels with multiple monitoring sites in them, and can that give you an idea of the surface level variability within one pixel? You say that the spatial matching ensures same geographical conditions, but how much can conditions vary within the TROPOMI pixel?

In our study, we extracted  $NO_2$  values from the Sentinel-5P dataset using Google Earth Engine (GEE). The data are provided as a Level 3 product on a regular 0.01° × 0.01° grid, corresponding to approximately 1.1 × 0.85 km over Spain. Due to the relatively high resolution and spatial distribution of monitoring stations, in our data, no more than one station is located within the same pixel.

However, in datasets downloaded directly from the Copernicus platform (Level 2 data), where the spatial resolution is close to  $3.5 \times 5.5$  km, it is possible to obtain multiple monitoring stations within the same satellite pixel. In these cases, the presence of multiple stations per pixel can help explore within-pixel variability, revealing how surface  $NO_2$  concentrations can differ even within the same satellite measurement. For example, in Catalonia, a single pixel could include both densely urbanized areas (e.g., Barcelona city) and adjacent natural zones (e.g., the Collserola mountain range), leading to substantial differences in surface  $NO_2$  concentrations. In contrast, rural zones tend to show less variability within a single pixel. However, due to the more dispersed distribution of stations in these areas, it remains uncommon to find multiple monitoring sites within the same pixel, even using Level 2 data.

We have added further explanation in Section 3.4.1, lines 213 to 219.

- Temporal matching: What is the time resolution of the surface level data? Are you matching only one timepoint, or averaging over a period of time? If averaging, what is the time interval you are taking coincident with the satellite overpass?

Surface-level data are recorded hourly. We conducted analyses using various averaging periods (e.g., 10:00 to 13:00, 11:00 to 13:00, 11:00 to 14:00, 12:00 to 13:00 and 12:00 to 14:00) to assess whether they improved the correlation with the satellite data. However, since these did not improve the correlation, we chose to present the results only for hourly surface-level values.

We have added further explanation in Section 3.4.1, lines 223 to 225.

**- Tables 1-3 could be combined.**

We have combined Tables 1-3.

**- Line 250: What about the BMA Zone? The correlation does not peak at 13:00 UTC**

The Barcelona Metropolitan Area (BMA) presents a more complex pattern compared to other zones, and in both years the correlation does not peak at the expected time of the satellite's overpass (13:00 UTC). Although a secondary peak is observed around 12:00 in 2022 and at 13:00 in 2023, the main peak appears around 19:00. In the remaining subgroups, this pattern is reversed: the maximum occurs around 13:00, while the 19:00 peak appears as secondary—except in the rural zone, where no secondary peaks are observed.

The particularity of the BMA zone could be influenced by evening traffic emissions, typical of urban areas with high population density and mobility, as well as by possible meteorological stability at that time, which coincides with what was observed: a more stable correlation during the coldest periods (although this hypothesis would require verification with hourly satellite data, which is not possible with the current satellite data available).

We recognize this as a distinctive feature of the BMA zone, which highlights its analytical complexity and the need for more detailed studies to further investigate this difference.

We have added further explanation in Section 4, lines 276 to 278.

- Figure 3 is misleading – it appears that you have an overpass every hour. It should be clear that this figure highlights the variability in surface NO2 throughout the day, and the TROPOMI only captures one point in that day.

We have updated the caption of Figure 3 to clarify this observation.

- Figures 3 and 4 should be combined with each subplot having two traces, color-coded by year.

We have combined Figures 3 and 4.

- Line 273: What assumptions did you make in converting columns to concentrations? This contradicts the initial paragraph which characterizes the spatial differences in satellite and insitu measurements.

The conversion aimed to quantify how Sentinel measurements change for each additional microgram measured at the surface and to assess potential measurement bias. The analysis focused primarily on the slope relationship rather than on absolute values. The conversion was used only to express both variables in the same units ( $\mu g/m^3$ ), which facilitates interpretation.

We have added further explanation in Section 4.1, lines 301 and 302.

- Line 377: Is a change in r from 0.43 to 0.44 interpretable as an increase?

The discussion has been rephrased on Section 5.2.1, lines 405 to 407.

- What do you mean by the complexity of urban/BMA areas? What causes this and what are the implications for surface  $NO_2$  levels in comparison to the tropospheric column? What makes this different than just having high surface  $NO_2$ ?

The main problem we may find in cities is the traffic variability. Within a  $3.5 \times 5.5$  km radius, we can find many types of neighborhoods (from traffic-calmed areas to high-density zones), which leads to significant variability in the measurements. This must be considered when estimating actual emissions, as we cannot extrapolate a single point measurement from one station to an entire area. In contrast, in rural or suburban zones, emissions tend to be more homogeneous, making it more feasible to correlate surface  $NO_2$  concentration with the tropospheric column.

We have added further explanation in Section 5.2.2, lines 427 to 429.

- Line 432: requires citation. What about ozone photochemistry?

The citation is in Section 5.4, lines 460 and 461.

- Section 6. Parts of the discussion and conclusion feel very redundant.

We have revised these parts of the document to avoid this redundancy.

**Reviewer 2**

**- 72-75: This part could be moved to methods in 3.3**

This part was removed, as suggested by the first reviewer.

- 76-82: This part could be moved to the introduction.

This part has been moved to Section 1, lines 49 to 55 of the revised manuscript.

**- 79: When presenting TROPOMI for the first time, it should be said that NO2 is one of the species measured.**

We have applied this suggestion in Section 1, lines 46 and 47.

**- 135, Section 3.2: What is the associated error of the measurements?**

Thank you for the comment. Ground-based  $NO_2$  measurements from the XVPCA network of the Generalitat de Catalunya are conducted following the European Standard EN 14211:2012, which defines the chemiluminescence method for measuring  $NO_2$  and  $NO_x$  concentrations (Gencat, 2025). The adoption of this reference method ensures typical uncertainties of around ±10% under standard operating conditions (CEN, 2012). We have modified the manuscript to include this information.

We have added further explanation in Section 3.2, lines 144 to 146.

**- 149, Section 3.3: Give details on the spatial resolution and coverage of the satellite pixels. Can you add information on the associated error of the measurements?**

Datasets downloaded directly from the Copernicus platform (Level 2 data) have a spatial resolution close to  $3.5 \times 5.5 \text{ km}^2$  (ESA, 2022). Based on mission requirements and validation reports, the random precision of the tropospheric  $NO_2$  column is  $0.7 \times 10^{15}$  molec cm-2 (1- $\sigma$ ), while the systematic uncertainty (bias) is between 25% and 50% (ESA, 2015).

In our study, we extracted  $NO_2$  values from the Sentinel-5P Level 3 dataset using Google Earth Engine (GEE). These data are provided as a Level 3 product on a 0.01° × 0.01° regular grid (~1.1 × 0.85 km over Spain). The L3 files in GEE are generated by applying harpconvert to the L2 product, after filtering out pixels with qa\_value < 0.75 (GEE, 2019). The process only remaps the data to a 0.01° grid and does not recalculate or propagate the precision band, therefore, it is adopted the official L2 uncertainty.

We have added further explanation in Section 3.3, lines 167 to 174.

- 160-163: Negative satellite values are discussed and observed in various scatter plots. The negative concentrations are explained as a result of the low NO2 levels and the instrument's noise exceeding the actual signal. Also, negative values in complex environmental conditions are mentioned. Can the authors further explain the reasons for the latter?

The tropospheric  $NO_2$  columns are retrieved using techniques that measure the absorption of sunlight by atmospheric  $NO_2$  molecules (ESA, 2022; Veefkind et al., 2012). Specifically, this retrieval involves isolating the tropospheric component by subtracting the estimated stratospheric contribution from the total  $NO_2$  column obtained through Differential Optical Absorption

Spectroscopy (DOAS). Due to uncertainties and assumptions inherent in this separation, particularly under conditions of very low  $NO_2$  concentrations, negative values can occasionally result. Several studies have documented and discussed this issue, including Jimenez and Brovelli (2023), Finch et al. (2022), and Douros et al. (2023), highlighting that these negative values are primarily retrieval artifacts or instrument noise rather than physically meaningful concentrations (lines 159-165). In practice, such negative columns tend to appear over regions with minimal  $NO_2$  or complex atmospheric conditions and remain within the instrument's typical uncertainty range.

We have added further explanation in Section 3.3, lines 176 to 183.

- Figure 2. Colour scale is qualitative (low-high). Please change it by quantitative values.

We have updated Figure 2 to include the actual column values and the corresponding units.

- 196: It is not necessary to extend so much the explanation on how to match UTC and CET time or explain what UTC and CET times are. Instead, some information is missing: what is the sampling frequency of satellite and AQ data? Are they averaged (e.g., to 1 hour) for matching? If a satellite pixel covers several AQ monitoring stations, even from a different typology, how are the data treated?

Surface-level  $NO_2$  is reported at an hourly resolution, while Sentinel-5P provides one observation per day over our study area, around 12:30 UTC. Regarding averaging values, we tested different hourly time windows (e.g., 10:00 to 13:00, 11:00 to 13:00, 11:00 to 14:00, 12:00 to 13:00, and 12:00 to 14:00) to evaluate whether this would improve the correlation between ground-based and satellite data. However, since these tests did not lead to a significant improvement in the data correlation, we chose to present the results only for hourly surface-level values.

In datasets downloaded directly from the Copernicus platform (Level 2 data), with a spatial resolution of approximately  $3.5 \times 5.5$  km (ESA, 2022), it is possible to obtain multiple monitoring stations within the same satellite pixel. In these cases, the presence of multiple stations per pixel can help explore within-pixel variability, showing how surface  $NO_2$  concentrations may differ even within a single satellite measurement. However, as in our study we have used the GEE Level 3 product on a  $0.01^{\circ} \times 0.01^{\circ}$  grid (~1.1 × 0.85 km) resolution (GEE, 2019), no multiple stations were found within the same pixel due to the dispersed distribution of monitoring sites in the study area.

Still, considering this more generally or in other study areas with more closely located stations, if such a situation occurred, the different station values would be associated with the same satellite pixel, and the analysis would still be carried out based on the individual station data.

We have added further explanation in Section 3.4.1, lines 213 to 219

- Figures 3 and 4 can be merged into a single 3x3 figure, overlapping data for 2022 and 2023.

We have combined Figures 3 and 4.

- Table 4: Mention in the table caption that the data corresponds to 13:00 UTC (it is not daily data).

We have updated the caption of Table 4.

- 271: Satellite measurements have been converted from mol/m2 to ug/m3, even though lines 255-259 do not recommend it, as it requires assumptions that influence the conversion and the comparability with ground concentration. The thread sounds confusing and needs to be rephrased. Also, there is no information on how the calculation has been done or which assumptions have been made.

We agree that converting satellite-derived tropospheric  $NO_2$  columns from mol/m² to  $\mu g/m^3$  involves complex assumptions that can affect comparability with surface-level concentrations. In our case, the conversion was performed using an average vertical profile assumption, with  $NO_2$  uniformly distributed within a 1 km thick boundary layer, and assuming standard atmospheric conditions. This approach was used only to express both variables in the same units ( $\mu g/m^3$ ) and facilitate interpretation.

We have rewritten the paragraph for clarity and included the calculation assumptions in Section 4.1, lines 301 and 302.

- Figures 6,7 and 8 could be grouped into 3x2 subplots. The same with 9 and 10; and with 11,12 and 13. Even some grouped figures could go to the supplementary info.

We have combined the Figures as suggested. The 13 Figures in the preprint document are now 7 Figures.

- 250-252: should be moved to section 3.4.1. The title could be Data preprocessing or similar, rather than Data matching, to describe the data preparation.

The suggestions have been made in Section 3.4.1, line 209.

- 308: This type of text is repeated many times in the text. Avoid this redundancy.

We have revised these parts of the document to avoid this redundancy.

- 337, Section 5: In general, in this section there are mentions of other studies. Please, give quantitative results found in those other studies for comparison with this work in each subsection.

We have revised Section 5 to include the quantitative results from the cited studies, as recommended. The values are in lines 381 and 384.

- 382: Regarding the scatter plots, the one of rural zones shows a markedly different distribution compared to urban and suburban zones, rather than urban zones being different from suburban and rural zones as mentioned in the text.

We agree that the scatter plots for rural stations present a cloud of points clearly concentrated at low values, which distinguishes them from those for urban and suburban areas. Our paragraph was intended to highlight another distinctive aspect: in the urban scatter plots, there are very few points around zero, while these are evident in suburban and rural areas. Thus, these zones exhibit different patterns, but for opposite reasons:

- Rural areas show most data clustered at the lower end of the range.
- Urban areas maintain consistently high NO2 levels and, therefore, few points around zero.

We have added further explanation in Section 5.2.1, lines 413 to 418.

**- 409: Is there any previous study using this classification?**

The typology-based classification used in our study follows the standard station classification adopted by the European Environment Agency (EEA, 2024). This approach has also been applied in recent studies using ground-based and Sentinel-5P data across different regions of Europe. For example, Pseftogkas et al. (2022) employed a typology distinguishing between seven categories (urban traffic, suburban traffic, urban background, suburban background, rural background, suburban industrial, and rural industrial) to analyse the agreement between TROPOMI data and ground-based NO2 measurements in Central Europe. Similarly, Morillas et al. (2024) grouped stations in Madrid by area type (urban, suburban, and rural) and further analysed station typologies within urban and industrial areas, finding subtle differences in the comparisons. Furthermore, Fania et al. (2024) applied a machine learning approach to estimate surface NO2 concentrations in Italy, incorporating typology classifications as key predictive features.

**- 432: Add a bibliographic reference.**

The citation is in Section 5.4, lines 460 and 461

- 432-438: This section discusses the seasonal dependence, but the text discusses how radiation varies within the day. The argument of more radiation, more photochemical activity and a reduction in NO2 concentration should be used considering the seasons, not the time of the day.

Thank you for the comment. We have revised this section to focus on the seasonal influence of solar radiation and photochemical activity, rather than on daily variation. The updated text explains that increased solar radiation in spring and summer enhances photolysis and other photochemical processes, reducing the lifetime of atmospheric NO2. This contributes to the lower correlations observed between satellite and ground-based measurements during these seasons.

We have revised Section 5.4 to attend the suggestion.

**Section 5: Overall, the authors should discuss the results considering:**

- How the diversity of conditions across a satellite pixel can affect the correlation (e.g. different zones or land use covered by the same pixel, or more or less dispersed emission sources in suburban and urban zones, etc).

The diversity of conditions within a pixel—such as differences in land use, emission intensity, or urban structure—can affect the correlation between satellite and surface measurements. For example, in Catalonia, a single Level 2 pixel may encompass both densely urbanized areas (e.g., Barcelona city) and nearby natural regions (e.g., the Collserola mountain range), leading to substantial variation in surface  $NO_2$  concentrations that is not fully captured by the averaged satellite value. This intra-pixel heterogeneity can reduce the observed correlation. In contrast, rural areas generally show less variability within a pixel, but due to the sparse distribution of stations, it is still uncommon to find multiple stations per pixel—even when using Level 2 data.

We have added further explanation in Section 5.5, lines 487 to 493.

-How comparable are ground NO2 levels to those reported by the tropospheric column, i.e. how well mixed is NO2 within the vertical column depending on the conditions, and which conditions favor that the main NO2 contribution to the vertical column is at the ground level.

In general, the tropospheric  $NO_2$  column is dominated by near-surface emissions rather than by chemical transformation, but meteorological conditions can influence both the vertical and horizontal dispersion of the pollutant. As a result,  $NO_2$  can be redistributed within the atmospheric column, potentially weakening the correlation between satellite-derived columns and ground-level concentrations.

We have added further explanation in Section 5.5, lines 494 to 501.

- (Point related to the previous one). Explain better what atmospheric stability means (e.g. line 473), i.e. how it impacts on how NO2 is mixed across the vertical column or accumulated at the ground level. This is especially relevant when discussing the seasonal classification. Discuss also if and how the height of the boundary layer may change, affecting the level of dilution of NO2.

We have improved our discussion to better highlight the important role that atmospheric stability plays in the vertical distribution and surface accumulation of  $NO_2$ . Not only temperature, but also the behavior of air masses, influences the dispersion of pollutants. Further explanation is in Section 5.5, lines 494 to 501.

**REFERENCES**

Copernicus. OBSERVER: Sentinel-4 – A new era in European air quality monitoring, *Copernicus*. Available at: https://www.copernicus.eu/en/news/news/observer-sentinel-4-new-era-european-air-quality-monitoring (last access: 2 April 2025), 2025.

Douros, J., Eskes, H., van Geffen, J., et al.: Comparing Sentinel-5P TROPOMI NO2 column observations with the CAMS regional air quality ensemble, *Geosci. Model Dev.*, 16(2), 509–534, doi: https://doi.org/10.5194/gmd-16-509-2023, 2023.

Eskes, H. J., Boersma, K. F., van Geffen, J. H. G. M., et al.: Product Specification Document for the TROPOMI NO2 Vertical Column. *Royal Netherlands Meteorological Institute* (KNMI), 2019.

European Committee for Standardization (CEN): EN 14211:2012 Ambient air – Standard method for the measurement of the concentration of nitrogen dioxide and nitric oxide by chemiluminescence, CEN, Brussels, 2012.

European Environment Agency (EEA): Monitoring station classifications and criteria for including them in the EEA's assessment products, EEA, Available at: <a href="https://www.eea.europa.eu/en/topics/in-depth/air-pollution/monitoring-station-classifications-and-criteria">https://www.eea.europa.eu/en/topics/in-depth/air-pollution/monitoring-station-classifications-and-criteria</a> (last access: April 2025), 2024.

European Space Agency (ESA): S5P Level-1b and Level-2 Product Validation Requirements Document, ESA, Available at: <a href="https://sentinel.esa.int/documents/247904/0/S5P-Level-1b-L2-">https://sentinel.esa.int/documents/247904/0/S5P-Level-1b-L2-</a>

numbered-validation-requirements.pdf/c8d27d16-9c48-4eff-8aa6-968f553b1fb9 (last access: April 2025), 2015.

European Space Agency (ESA): Sentinel-5P Level 2 Product User Manual Nitrogen Dioxide, ESA, Available at: <a href="https://sentinel.esa.int/documents/247904/2474726/Sentinel-5P-Level-2-Product-User-Manual-Nitrogen-Dioxide.pdf">https://sentinel.esa.int/documents/247904/2474726/Sentinel-5P-Level-2-Product-User-Manual-Nitrogen-Dioxide.pdf</a> (last access: April 2025), 2022.

Fania, A., Monaco, A., Pantaleo, E., Maggipinto, T., Bellantuono, L., Cilli, R., Lacalamita, A., La Rocca, M., Tangaro, S., Amoroso, N., and Bellotti, R.: Estimation of Daily Ground-Level Air Pollution in Italian Municipalities with Machine Learning Models Using Sentinel-5P and ERA5 Data, Remote Sens., 16(7), 1206, https://doi.org/10.3390/rs16071206, 2024.

Finch, D. P., Palmer, P. I., and Zhang, T.: Automated detection of atmospheric NO2 plumes from satellite data: a tool to help infer anthropogenic combustion emissions, *Atmos. Meas. Tech.*, 15(3), 721–733, doi: <a href="https://doi.org/10.5194/amt-15-721-2022">https://doi.org/10.5194/amt-15-721-2022</a>, 2022.

Gencat: Xarxa de Vigilància i Previsió de la Contaminació Atmosfèrica (XVPCA) – Conceptes clau, Gencat, Available at:

https://mediambient.gencat.cat/ca/05\_ambits\_dactuacio/atmosfera/qualitat\_de\_laire/avaluacio/xarxa\_de\_vigilancia\_i\_previsio\_de\_la\_contaminacio\_atmosferica\_xvpca/conceptes\_clau/ (last access: 2 April 2025), 2025

Google Earth Engine: COPERNICUS/S5P OFFL L3 NO2: Tropospheric NO2 (Offline, Level 3), Available at: <a href="https://developers.google.com/earth-engine/datasets/catalog/COPERNICUS">https://developers.google.com/earth-engine/datasets/catalog/COPERNICUS S5P OFFL L3 NO2 (last access: April 2025), 2019.</a>

Jimenez, J., and Brovelli, M. A.: NO2 Concentration Estimation at Urban Ground Level by Integrating Sentinel 5P Data and ERA5 Using Machine Learning: The Milan (Italy) Case Study, *Remote Sens.*, 15(22), 5400, doi: https://doi.org/10.3390/rs15225400, 2023.

Kim, J., Jeong, U., Ahn, M. H., et al.: New era of air quality monitoring from space: Geostationary Environment Monitoring Spectrometer (GEMS). *Bulletin of the American Meteorological Society*, 101(1), E1–E22. https://doi.org/10.1175/BAMS-D-18-0013.1,2020.

Landgraf, J., aan de Brugh, J., van Kempen, T., et al.: S5P/TROPOMI ATBD of the Total and Tropospheric NO2 Data Products. Sentinel-5 Precursor/TROPOMI Level-2 Algorithm Theoretical Basis Document (ATBD), 2018.

Morillas, C., Martínez, S., and Sobrino, J. A.: Analysis of the Relationship between TROPOMI NO2 Observations and Ground-Based Measurements in the Madrid Metropolitan Area, Remote Sens. Appl. Soc. Environ., 33, 101083, <a href="https://doi.org/10.1016/j.rsase.2023.101083">https://doi.org/10.1016/j.rsase.2023.101083</a>, 2024.

Petetin, H., Guevara, M., Compernolle, S., Bowdalo, D., Bretonniere, P. A., Enciso, S., Jorba, O., Lopez, F., Soret, A., and Perez Garcia-Pando, C.: Potential of TROPOMI for understanding spatio-temporal variations in surface NO2 and their dependencies upon land use over the Iberian Peninsula, *Atmos. Chem. Phys.*, 23(7), 3905–3935, doi: https://doi.org/10.5194/acp-23-3905-2023, 2023.

Pseftogkas, A., Koukouli, M.-E., Segers, A., Manders, A., van Geffen, J., Balis, D., Meleti, C., Stavrakou, T., and Eskes, H.: Comparison of S5P/TROPOMI Inferred  $NO_2$  Surface Concentrations with In Situ Measurements over Central Europe, Remote Sens., 14(19), 4886, https://doi.org/10.3390/rs14194886, 2022.

Veefkind, J. P., Aben, I., McMullan, K., et al.: TROPOMI on the ESA Sentinel-5 Precursor: A GMES mission for global observations of atmospheric composition for climate, air quality, and ozone layer applications, *Remote Sens. Environ.*, 120, 70–83, doi: <a href="https://doi.org/10.1016/j.rse.2011.09.027">https://doi.org/10.1016/j.rse.2011.09.027</a>, 2012.

Zoogman, P., Liu, X., Suleiman, R. M., et al.: Tropospheric Emissions: Monitoring of Pollution (TEMPO). *Journal of Quantitative Spectroscopy and Radiative Transfer*, 186, 17–39. https://doi.org/10.1016/j.jqsrt.2016.05.008, 2017.